# Grief (Work) Is Heart (Work): A Critical Race Feminista Epistolary Exchange as an Offering on Death, Grief, and Well-Being to Academia

**Nichole Margarita Garcia** [1,*]  **and Solano F. Garcia** [2]

1 Educational Psychology and Higher Education Program, Graduate School of Education, Rutgers, The State University of New Jersey, New Brunswick, NJ 08901, USA

2 Independent Researcher, New Brunswick, NJ 08901, USA

* Correspondence: nichole.garcia@gse.rutgers.edu

**Abstract:** This article centers around my work as a critical race feminista; an academic experiencing consistent attacks on the scholarship I produce while also being a tía (aunt), an active griever, and a godmother to my eldest nephew, Solano Garcia. This is the first time that my nephew and I will have shared our most private papelitos guardados (intimate guarded papers). In this article, we respond to the paucity of Black, Indigenous, and People of Color-centered death, grief, and well-being in academia. Using a critical race feminist epistolary methodology, we document our epistolary exchanges that contain dehumanizing attempts on our bodymindspirit matrices as active grievers of color confronting the premature death of my brother, who died at the age of 37 in the summer of 2021. Unlike the 'western' psychotherapeutic tradition of overcoming death and grief, we stake a claim, sit with it, and affirm it as an ongoing process. We argue that recognizing and affirming death and grief is a life-making process that creates spaces for healing through our epistolary offerings. This article aims to offer BIPOC faculty, staff, students, and their families life-affirming strategies towards radical self-care, love, and intergenerational collective healing within a sociopolitical context that operates as a surveillance mechanism.

**Keywords:** death; grief; critical race feminista epistolary methodology; academia well-being

## 1. Introduction

Contemplating death has always been a subject that leads me back to love. Significantly, I began to think more about the meaning love as I witnessed the deaths of many friends, comrades, and acquaintances, many of them dying young and unexpectedly [1] (p. xxii).

Making sense of the severity of the COVID-19 pandemic and the aggressive attacks on critical race theory as a critical race feminista has created much conflict and tension for myself and many others. As with many critical race theorists, our lived realities are not often divorced from the research we engage in; instead, they inform our work and how we do it. The attacks on critical race theory have swept through the nation, including the passing of legislation banning it from K-20+ public education. While our work is under attack, our physical and mental well-being also faces persecution at the hands of racism, white supremacy, and colonial logic. The COVID-19 pandemic immediately brought to fruition public health, educational disparities, and inequities among Black, Indigenous, and People of Color communities (BIPOC). The impact of COVID-19 on the BIPOC communities has garnered much concern as infections and deaths are at disproportionate rates [2]. For BIPOC faculty, staff, students, and their families, the dire consequences of a global shutdown would result in inaccessible health care, economic loss, insufficient education, premature death and death, and an increase in vulnerability for populations already experiencing structural violence and injustice. Historically, institutions of higher education (IHE) have been sites

where BIPOC scholars engaged in critical race theory as a field and praxis are harmed and stressed by everyday racism, sexism, and classism. Still, when coupled with the pandemic, these instances are only further exacerbated.

As an academic community, we have swept death, grief, and well-being under the rug as if it did not warrant our time to process, mourn, and heal. On 21 January 2022, I wrote a very public and intimate blog for *Diverse Issues in Higher Education* titled "Haunted by Grief". In that short piece, I asked the academic community how death and grief could be humanized in a way that did not dismiss racial violence, capitalist regimes, and a global pandemic. I asked, "Have you taken the time to honor your grief?" This was in response to bearing witness to the premature death of my brother, Richard Jr. Garcia, who died at 37 in the summer of 2021 from a massive heart attack and after a life-long battle against bipolar disorder.

This article centers around my work as a critical race feminista; an academic experiencing consistent attacks on the scholarship I produce while also being a tía, an active griever, and a godmother to my eldest nephew Solano F. Garcia. This is the first time my nephew and I will have shared our most private papelitos guardados [3]. We are pulling out our papelitos guardados "for review and analysis, to speak out and share them with others" [3] (p. 1). In this article, we respond to the paucity of BIPOC-centered death, grief, and well-being in academia, not only as human beings but as critical race scholars conducting equity work and working to create a life-making space. Using a critical race feminista epistolary methodology [4,5], we document our epistolary exchanges that contain dehumanizing attempts on our bodymindspirits [6] as active grievers of color. Unlike the Western psychotherapeutic tradition of overcoming death and grief, we "stake a claim, sit with, and affirm it" [7] (p. 620). As with hooks [1], we return to love as a means to return to what has been lost in the physical world but lives on in our beating hearts and memories. We argue that recognizing and affirming death and grief is a life-making process that creates spaces for healing through our epistolary offerings [7].

## 2. Bereavement, Grief, and Well-Being in Academia

"Academia rewards those who can make hardship invisible, who can be productive amid and despite crisis" [8] (p. 1).

In this literature review, we provide the limited scholarship on bereavement, grief, and well-being in academia. Bereavement and grief are often used interchangeably but differ in meaning and experience. For this literature review, we define bereavement as the period when an individual first loses a significant other through death [9,10]. Grief is the emotional, cognitive, and behavioral response to loss [10]. Most studies conducted on bereavement, grief, and well-being in academia tend to focus on undergraduate students [10–14]. When studies focus on faculty, it is how faculty can best support and assist undergraduates in their bereavement and grief processes rather than being the individual who is bereaved [10–14]. Therefore, this literature review is limited in scope and depth.

Generally, bereavement in adulthood addresses the death of a spouse or parent, whereas less common is the loss of a child [10]. Thai and Moore [10] argue that the age of a bereaved individual may play a role in relation to when the death occurs and the need for support. The older an adult is, the more susceptible they are to the death of a loved one, such as a spouse, sibling, adult child, grandchild, and/or parent. Bereavement is a complicated process that potentially impacts an individual's "psychological, physiological, social, economic, and spiritual wellbeing" [10] (p. 10). While not exhaustive, the scholarship on bereavement and academia tends to focus on the needs, experiences, and sources of support for undergraduates (p. 10). For undergraduates who experience bereavement in academia, they are more vulnerable to psychological health consequences such as low mood, inability to sleep, loss of appetite, depression, anxiety, and academic consequences such as a lower GPA, dissociation from institutions, social and academic withdrawal, and higher attrition rates [11,15,16].

Despite numerous studies examining bereavement among undergraduates, little or no research has been conducted on bereaved faculty in academia. Fitzpatrick [17] conducted one of the few studies focusing on the experiences of bereaved faculty returning to the university workplace environment to document the strengths and weaknesses of their respective higher education institutions. Through in-depth interviews, Fitzpatrick [17] found four main findings that bereaved faculty experienced: health-related symptoms, employment strains and support, the university environment, and supportive gestures. Faculty described emotional and health-related symptoms such as sleeplessness, exhaustion, weight gain, anger, sadness, confusion, and strangeness. Returning to the university workplace, the faculty found it difficult to continue business as usual. Often, they had contradictory experiences. Faculty found some solace for a very limited amount of time, or the loss they had experienced went unacknowledged or was diminished by their colleagues and/or the university. Fitzpatrick [17] argues that bereavement among faculty is complex because "emotional reactions to grief and physical health symptoms go unrecognized due to the complexities and demands of the educator's role" (p. 83).

While there is limited scholarship on bereavement for students and faculty in higher education contexts, much research has focused on models of grief [18]. Kubler-Ross' [19] Stages of Grief model is the most widely known and asserts that individuals move sequentially through five stages: denial, anger, bargaining, depression, and acceptance. While this model is well-known and cited often, it does not consider the influence of culture, race, gender, or other markers of social identity or societal structures. This model also assumes that grief is an "overcoming" outcome delineated by stages, which may not be the case for everyone. Recently, the COVID-19 pandemic prompted grief to be considered as a model of "your world and the ball of grief" [20]. This model considers that individuals may never overcome their grief or its associated pain. Instead, while individuals navigate the world and their experiences increase, their grief might become smaller and more manageable. This model also considers that an individual can instantaneously experience multiple forms of grief. Grief is not limited to loss based on death, but rather, grief is associated with all types of loss, such as employment, identity, and/or significant life events [18,20].

Kumar [18] in a systematic review of the literature, identifies four levels of grief: grief for self, grief for the loss of a loved one (relational grief), collective grief, and ecological grief (p. 103). Grief for self refers to loss regarding life events such as milestones or identities. Relational grief is grief in the traditional sense of losing a loved one due to death. Within relational grief, there are different forms an individual can experience, such as acute, complicated, anticipatory, and/or disenfranchised grief. Acute grief focuses on the immediate response to loss, which is disruptive and extreme. In contrast, complicated grief is the prolonged response to loss, which results in a yearning and difficulty in acceptance [21]. Anticipatory grief is when an individual anticipates losing a loved one due to declining health conditions. Disenfranchised grief [22] refers to an individual or community not being afforded the right to grieve following a death or loss and not provided sympathy to grieve their loss. Collective grief accounts for a large amount of death that can occur in a community where grief is shared and communal—an example of this is the COVID-19 pandemic. Lastly, ecological grief is "grief felt in relation to experienced or anticipated ecological losses, including the loss of species, ecosystems, and meaningful landscapes due to acute or chronic environmental change" [23] (p. 540). As documented in the literature, grief is complex, varies, and is not limited by time or circumstance. Grief can and will be experienced differently by individual or collective communities, and it can have mental, emotional, and physical effects.

Overall, disciplines studying death, grief, and bereavement include, but are not limited to, sociology, nursing, psychology, philosophy, literature, history, and bioethics. Studies on bereavement and grief have often been grounded in positivism not attuned to individual or structural violence of race, racism, or white supremacy in BIPOC communities while experiencing bereavement and grief [24,25]. Doka [26] describes disenfranchised grief as "grief that results when a person experiences a significant loss and the resultant grief is not

openly acknowledged, socially validated, or publicly mourned" (p. 223). Over the past 30 years, disenfranchised grief has maintained significant applicability to communities that live on the margins of U.S. society and globally. Most recently, Turner and Stauffer [25] expanded disenfranchised grief to "suggest that non-death losses specifically related to discrimination, marginalization, and oppression comprise another unique typology for consideration in the context of disenfranchised grief" (p. 6). In an edited volume, Turner and Stauffer [25] underscore how disenfranchised grief is manifested between social identities (e.g., race, gender, sexuality) and the impact of discrimination, marginalization, and oppression. For individuals and communities that live on the margins of U.S. society, the combination of disenfranchised grief and systematic oppression has pervasive impacts.

In higher education scholarship, little to no literature addresses death, grief, and well-being in academia, especially for BIPOC faculty, staff, students, and their families. Despite this limitation, Nicolazzo [27] powerfully offers insights into grief as an academic. They state:

> "How we feel grief with/in higher education has often meant not feeling grief with/in higher education. In so doing, we create worlds in which we encourage others to not feel with/in higher education, which then gets passed along. Sagas are created around those of us who feel the altering effects of grief, and who share our grief openly. We are too much and our grief gets in the way of our work, which is always posed as separate from feeling". (p. 21)

Nicolazzo [27] makes it evident that in academia, those experiencing grief are not allowed to feel while working in these institutions. Confirming this belief, Harrison [8] eloquently reminds us that academia rewards individuals' abilities to create invisibility when experiencing death and grief while maintaining productivity. Therefore, this article addresses the scarcity of BIPOC-centered death, grief, and well-being research in academia. As critical race scholars, we seek to humanize death and grief as an ongoing process versus an overcoming while working to create a life-making space that is equity-centered.

## 3. Community Memory: A Critical Race Feminista Epistolary Methodology

Critical race theory (CRT) in education draws from several disciplines—including civil rights, ethnic studies, gender studies, and critical legal studies—to examine and transform the relationship between race, racism, and power [28,29]. It is motivated by social justice and characterized by activism to eliminate racism as part of a broader effort to end subordination on the lines of gender, class, sexual orientation, language, and national origin. Education scholars have adopted CRT to scrutinize how race, racism, and white supremacy permeate educational institutions, discourse, and practices [30]. We draw on the five tenets of critical race theory to situate ourselves as active grievers of color to disrupt Western' psychotherapeutic and pathologizing traditions of overcoming death and grief. Tenet 1: as active grievers of color, it is critical to center race, racism, and the intersections of other forms of oppression towards death and grief equity. To date, research on death and grief for BIPOC communities is grounded in positivism, not attuned to individual or structural violence of race, racism, or white supremacy for BIPOC communities experiencing death and grief. Tenet 2: as active grievers of color, we seek to challenge dominant ideologies and discourses of Eurocentric positivism that pathologizes death and grief for BIPOC communities. Death and grief are not isolated events, and there must be an engagement with systems and structures that fail to recognize power, privilege, and racial trauma when BIPOC communities experience loss. Tenet 3: as active grievers of color, we center our experiential knowledge to disrupt dominant ideologies regarding BIPOC death and grief. We do this by illuminating our responses to how and in the ways we grieve, combining with our experiences as People of Color who face individual and structural violence due to racism and white supremacy. Tenets 4 and 5: as active grievers of color, we draw from interdisciplinary perspectives to subvert power and privilege to extend grief paradigms towards social justice aims and action.

As with CRT, Chicana feminist scholars theorize from their lived experiences and use this knowledge to identify, challenge, and analyze multiple forms of oppression while acknowledging the intersections of race, gender, class, and/or sexuality that inform the experiences of Chicanx/a/o [31–33]. Chicana feminists in education have been foundational in theorizing from their lived experiences to give names to epistemologies, pedagogies, and methodologies to counteract Western/Eurocentric paradigms and discourses that dominate academia [32]. Chicana feminisms offer a "subalternized theoretical set of tools" to identify cultural knowledge, strategies of resistance, and modes of inquiry [34] (p. 466). One of these tools is acknowledging the Western/European mind/body split [6]. We use Chicana feminisms to resist the fragmentation of the bodymindspirits as an object of desire. Brown bodies are often othered in schooling and society due to white supremacy and colonial logic [6,35–37]. The suturing of our bodymindspiritsin our grieving processes is critical as it is frequently dismissed and unacknowledged. In this article, we reclaim our Brown bodies through our grief to collectively heal our bodymindspiritstowards well-being [6,37].

Therefore, we theoretically braid critical race theory and Chicana feminist theories to employ a critical race feminista methodology [35,36]. A critical race feminista methodology is an anti-colonial praxis that braids theory, qualitative research methods, and critical consciousness. Delgado Bernal et al. [36] states:

> a critical race feminista methodology is one that seeks to disrupt traditional research paradigms that (re)produce systemic oppression through an intentional praxis to build and create theoretical and methodological strategies that sustain our humanity and honor those with whom we engage in the research process. (p. 119)

In employing a critical race feminista methodology, Escobedo and Camargo Gonzalez [5] propose an epistolary method, which "creates opportunities for Women of Color to challenge the historical legacies and contemporary manifestations of racist and gendered oppression through the construction of handwritten and digital letters addressed for, though not necessarily delivered to relatives, institutional agents, future generations, and the self" (p. 265). Much the same as platícas, a critical race feminista epistolary methodology involves intimate exchanges that are documented in the form of letter writing and have the potential to heal. Cisneros [38] argues that an epistolary method as an act of writing is a refusal of white supremacy and colonial logic. When guided by critical race feminista theories, an epistolary method acts as a familial archive that strengthens the sense of self and belonging in community contexts [38].

We engage a critical race feminista epistolary methodology through our intergenerational epistolary exchanges by documenting our knowledge, skills, and abilities pertraining to death, grief, and well-being in academia and beyond. Before this experience, we had always maintained a close bond. I left home to pursue graduate studies when Solano was five years old and taught him the art of letter writing to communicate long-distance. We have carried on this tradition throughout the years, mainly when Solano was coming of age. As Solano is a young adult, we have returned to our letter writing to heal while grieving collectively. We began our epistolary exchanges during the summer of 2023 as we prepared college applications for Solano. A personal statement is required as part of the college application. In preparing Solano to write his statement through free writing, I asked him what he wanted to write about. He eloquently took a deep breath and said, "Death". I responded with, "And what about grief?" He responded, "Yeah, that too". I suggested we use letters to put words on the page. From there, our epistolary exchanges were birthed.

Our epistolary exchanges are between me (Nichole, age 35) and my nephew (Solano, age 18), are situated chronologically, and occurred over the course of a few months from July 2023 to October 2023 through email correspondence (i.e., digital letters). As active grievers of color, our epistolary exchanges reflect on our grieving process marked by the premature death of my brother. Richard Jr. Garcia who passed away from a massive heart attack after working a night shift at the age of 37 in the summer of 2021. Our epistolary exchanges transpired organically, with time in between to process what each of us wrote emotionally, physically, and mentally. We want to bring to light that both of us during

this time were attending and continue to attend grief therapy. We also want to clarify that this is only a fragment of our exchanges, as not all that we wrote needs to be witnessed, documented, or shared with the public or academia. We begin our epistolary exchanges with a prologue, breathing life back into my brother's obituary and handwritten words, which we discovered among his personal possessions.

## 4. Prologue: The First Day of the Rest of My Life

Richard Jr. Garcia a high school graduate of Judge Memorial Catholic High School and a college student at the University of Utah created many friendships. Richard was best known for his loving heart, charming smile, foolhardy spirit, generosity, and love for sports. As a die-hard Raider Nation fan, without a doubt, you could hear his cheers a mile away. He enjoyed comedy, especially Ridiculousness and Ghost Hunters, which he loved to watch with his dad and sons. He bordered on hoarding/collecting random knick-knacks and making people laugh with his infectious belly laughs. His most cherished times were spent with his sons, whom he loved endlessly and who were his ultimate pride and joy and reason for living.

While cleaning out his belongings with Solano, we found the only piece left of his handwriting. It was a one-page letter in a marble-wide ruled notebook to his sons. The letter concerned his battle with bipolar disorder written while having been admitted to a psychiatric ward for attempting to end his life. This life-pausing event took place a week before my dissertation proposal defense. Solano was ten years old. What harm had racial violence inflicted on his bodymindspirits that he would want to fulfill the reality of premature death? We begin our epistolary exchange with the last handwritten words of Richard because he had committed to becoming well. Yet, he was met with systems (i.e., schooling, health care, mental health) that would always refuse him. However, time is serendipitous, and we found ourselves writing to each other on the same exact day he had written this entry nine years prior. In a letter to his sons, he stated the following:

17 October 2014

The first day of the rest of my life, or should I say the end of my old life and the beginning of my new life. How did I end up here? I sure hope one day I find out what's wrong with me. I have a long road ahead of me, but one thing is for sure: when I put my mind to something, I give it 110%. I know I can beat this. I don't have a choice because I have two little boys that need me just as I need them.

## 5. Memory Keepers: An Epistolary Exchange

Strengthening our relationship with grief means making a friend of loss. You may ask, "why on earth would anyone want to be friend with the pain of loss?" I suggest it because it is an inevitable part of life, and to be enemies with loss is to reject part of ourselves. The pain of loss is real, and yet we desire to sweep it away and 'move on' from it  [39] (p. vii).

Dear Tía,                                                                                                    29 July 2023

No matter how many male role models are in one's life, none of them will ever be your father. It is fascinating the types of things you discover once you have grief play a role in your life. For me, losing my father not only changed my view on the world around me but also made me understand different things about my life and the impact that loss has on one moving forward. I lost my father when I was 15 and found out about all the lies, we had been told when I was young. One thing that I found not to be true is that "time heals all". Time does make the pain hurt less, but grief never truly goes away. Every milestone I have gone through in my life hurts me because I do not have the fatherly approval that I yearn for as a young adolescent. Grief does bear a lot of pain, but it teaches you one thing that not many people understand. Grief gives you the gift of resilience. Every time my father is not there it breaks my heart but because of this experience I am able to push through the agony and excel in whatever challenge comes before me. Losing my father gave me the ability to understand that if I can get through the pain of true loss, I can do anything that I put my mind to.

A big part of grief is how you go through the stages. The five stages of grief are denial, anger, bargaining, depression, and acceptance. Throughout the grieving process, these five emotions may be very prevalent in your life. The stages go in no certain order or time period, the emotions come and go as they please and the experience of these emotions is different for everybody. The biggest emotion I felt through my grief was anger. I was angry because everything felt out of my control. It felt like I was going through the motions instead of living. It upset me because I wanted to enjoy life again like I did when my father was around. I found out that no matter what I tried or how angry I was, life was never going to be the same as it once was. Now I had to start from scratch, I could no longer rely on old things to make me happy, I had to find new things to take interest in and new passions to start enjoying life again.

Love,
Solano

Dear Solano,                                                                                        15 August 2023

There is not a day that I do not awake to the thought of you and your brother. I feel the deep pain in my own heart of what it meant to lose my brother, your father, and that pain is only intensified when I think of you two. Looking back, I came home in March 2021 during COVID-19 to receive a vaccination as they were not in abundance in the tri-state area. I did not know that those few months I was home would be the last moments I would have with your father. I remember being very concerned when I first got home because your dad had gained weight, and I knew he was struggling with his bipolar disorder. I left to go back east in June 2021 to return home (Utah) for a year while my university went fully remote for fall 2021, and I would be entering a sabbatical (research leave) in spring 2022. Just three days after my return, our lives would drastically change forever. It broke my heart when I was the one who had to tell you and your brother that your dad had passed away. However, I believe the universe works in mysterious ways, as I was able to be with you and our family for an entire year in the aftermath of your father's death.

I had initially set out to write this piece on the grief of my own grief during tumultuous times while integrating data I had collected during COVID-19 on Latinx/a/o college students. Like the students I interviewed, we, too, were devastated by not just the pandemic but the aftermath of what it meant for our communities' mental, emotional, and physical well-being. The students I interviewed discussed experiencing racism, lack of health care, and experiencing basic needs insecurities. Like them, we lost many family members, and now we are active grievers of color, but I want you to know WE are not alone.

To provide more context, this article aims to discuss how scholars like me who identify as critical race theorists work on racial diversity and inclusion in schools and humanizing wellness while under attack. While grieving alongside you, Grandma, and Grandpa, I still had to show up for work as a Critical Race Feminista scholar. At the same time, legislation across the nation was being introduced to ban Critical Race Theory in public education. While in Utah, S.R. 901 Senate Resolution on Critical Race Theory in Public Education was introduced into the state legislation and is still ongoing.

It is more important than ever to consider how equity work is informed by racial stress, trauma, and cultural taxation (Kohli & Pizarro 2022) and question how equity work impacts mental and physical wellness. I hope, and with your help, that we can center grief work as heart(work) and offer strategies that we have used in our own relationships to resist and live during the current sociopolitical context.

<div style="text-align:right">

Love,
Tía

</div>

Dear Tía,                                                                        8 September 2023

Grief is such an obscure topic to many people of my age. Having lost my father at the age of 15, it made me feel like I had to grow up so much faster than my peers. After the passing of my father, I had to be strong and take care of things on my own. Having you in Utah for that year was the best thing that could have happened for me because I was not alone. You and I had very different relationships when it came to my father, but he loved both of us unconditionally. Even though he never said it to my face, I could always tell how proud my father was of his little sister doing big things in the world. When he passed, our family was devastated because we all knew this day would come, but none of us knew it would have been this soon.

I constantly saw Dad at his highest and lowest and knew him better than most people. He never kept anything from his kids, a blessing and a curse. It was nice to know who my father was, but at the same time, I still needed him to be mature and act like he was my parent. Before you came back to Utah, he had a very hard time that he did not share with the family. What people do not know is that right before he passed away, he struggled to get out of bed. It was hard to watch him, and there was not a single thing I could do about it. All I could do was hug him, knowing that no matter what I did, it would not fix what he had going on inside him. After he passed, I realized that it was his time to go. He was barring a heavy heart every single day.

Now, picking up two years later, I realize how much I have changed because of this experience. Grief toughened me up to deal with the pain that I bear. Grief taught me that most people who are mentally strong and stable have gone through so much hardship and obstacles to become strong. You are the perfect example of this Tía. You and I are like two peas in a pod because, just like me, you lost someone very important to you when you were 15 years old. You know what it is like navigating grief and still trying to grow up. We both had to grow up fast and keep pushing forward even when nothing made sense. I would love to know how you feel grief has affected you differently when you were a teenager versus now.

<div style="text-align:right">

Love,
Solano

</div>

Dear Solano,

10 September 2023

Sometimes, I wish the universe did not hand you such a raw deal so young. Yes, you are right. I learned about death, grief, and loss earlier than I had hoped I would. I would never wish this pain on my worst enemy because grief is not describable to anyone who has not experienced it. The most significant difference between when I was your age and now is that I sought help immediately after your father passed away. When my best friend passed away, I was not allowed to grieve publicly. Grandpa and Grandma never talked about her passing, and I was left alone to deal with my emotions, thoughts, and, most significantly, my loneliness. I entered a dark place, got involved with the wrong people, and got into major trouble. It took me about a year and a half to get back on the right path.

During my senior year of high school, I just wanted to be done with school to experience something different. I knew college would open doors for me, and I could meet new friends who had similar goals and wanted to be something in life. I knew I had to channel my pain into something that could create change and help others. I loved to read and write, and that is what I did. If you want some time, I can share my journals with you from those times to give you some perspective on what my 15–18-year-old brain was going through.

I also witnessed your father go through an ebb and flow of emotions from extreme highs to lows. I never fully understood why he would feel so deeply about things that made no sense to me. However, I admired it at the same time. He loved you and your brother so much; you both were his gifts in life. His reason to live and keep going. Your father loved you deeply, and I witnessed many heartbreaks he endured. I am afraid his heart gave out on him, but his love still lives through us.

It took me 15 years to finally be at peace with my best friend's death, and the year my heart was finally able to suture the open wound into a scar, your dad passed away. It was and still is as if I have not known a life without grief being a driving force in it. Can you imagine? The biggest difference between now and then is that I have had to learn to live with grief and joy simultaneously. I have had to learn that life, while hard and poses its obstacles, is also beautiful. Our grief has changed us forever, and we will have to be okay with becoming new versions of ourselves. We only have one life to live, and we must live it for ourselves and those who have left us. I know that when it is my time to go, I will be met with open arms and reunite with your dad but until then, I am an active griever on the mend. You bring me much joy, and I am very proud of the man you are becoming.

Love,
Tía

Dear Tía,

22 September 2023

Today, grief has me down. I woke up sick this morning with a cold and a headache, and it has not felt good. But being sick reminds me of a time when I was younger, and I had my father. When I would get sick as a kid, my mom would always drop me off at my dad's. He would make me some soup and take care of me. We would spend the day watching movies and resting. Dad knew how to kick back, that is for sure. I remember my father as a man who wanted to help but could not help himself. Now that I am sick again, I can only think about those times. Times that I wish I could get back. You never realize how vital the meaningless things are until they are gone.

When I am sick, I want to be completely isolated from the world. I do not want to talk to anyone, and I do not want to deal with anyone else's problems. I feel more comfortable when I am sick because my appearance finally reflects how I feel. Now I have a reason to stay in bed, now I have a reason to ignore the world, and now I can give myself time to heal. Grief is being sick on the inside; if you don't give yourself enough time to rest, you will stay sick. I am tired of feeling sick, tía. I am tired of struggling to be happy in a world that has taken so much from me. I have gone through so many life-changing events at such a young age that it feels like I have the weight of the world on my shoulders. But I guess if you talk to any other 18-year-old, they will say the same thing.

The times are hard, but we are all getting through it in different ways at different paces. It makes me wonder more about how grief affects you in your current life. In what way is grief affecting your life currently?

Love,
Solano
29 September 2023

Dear Solano,

I hope you feel better physically soon. I was happy to hear about the care your dad once gifted you while you were not feeling well. I can read and feel your love for one another lift off the page. I do not know what it means not to have a parent take care of me. I will learn much from you once Grandma and Grandpa transition from this physical world. Now, as an adult, when I get sick, I still yearn for my parents. I always text them, and they always respond with how to best take care of myself. I do not have words to provide you comfort, nor do I believe words in your deepest moments of grief would comfort you unless it were from your father.

Grief is an embodied experience with its own process of moving through the body on its own terms. Your perception of grief as a sickness is very insightful. As a society, we are not taught how to grieve publicly, especially for people your age. It is not something openly talked about, and as you have found out, it is not as relatable as happiness. Since your father died, grief has affected my life tremendously. As a Critical Race Feminista who writes about the effects of racial violence on BIPOC communities, I found it impossible to put my thoughts as a scholar on paper after your father passed away.

The fact that your father died very young or what they call premature death as a Latino male, sits with me heavily. Black and brown boys and men, like your brother and you, must be careful when navigating U.S. society as it is a dangerous space for you both. U.S. society is ingrained with racism and white supremacy. Premature death occurs often for Black and brown boys and men, and your father is among those who died before they were supposed to. This is the first time I am making public what your father was battling and how his heart gave out. It is no easy task for me because I know the research on racial and structural violence that leads to premature death. Before your father died, I wrote a public blog about him. He was truly upset with me. I believe he did not want people to know the demons he was fighting. I attempted to express to him that his battle with bipolar disorder was nothing to be ashamed of, and as his sister, I was learning more about it to cope best and to know how to have a relationship with him in our adulthood.

He told me he worried about what you and your brother would think of him. He asked me not to write about him again. I respected his wishes, and so I didn't. During COVID-19, on 8 March 2020, he sent me a text that read, "Hey sis, you can write about me at any time you feel", with thumbs-up emojis. I still have his text messages saved, and I return to this one a lot as I grieve, including this morning before I sat down to write you back. I want to honor his memories. I was reading and came across the phrase "memory keepers", which refers to when a sibling loses another sibling. I believe I am his memory keeper because I have the most memories with and of your father outside of Grandpa and Grandma.

To answer your question. Currently, grief is being written on many pages of my journal. I mostly write memories I have of your father so I can one day share them with you and your brother. I feel grief in my body every day. I have gained weight since he passed because I am experiencing depression and anxiety. My therapist says I am high functioning, but in all reality, I have to function to keep my job. You see, my job only rewards those who are productive and put on a happy face while you are slowly—to use your words—"sick" inside.

I think most BIPOC faculty are not allotted time to grieve or are not seen as worthy of taking the time to do so. Many of my colleagues were aware your father had passed, and no one said a word. Only a few women of color faculty did. I have witnessed my other colleagues lose loved ones. They were given flowers, sympathy emails, and cards and provided time and space to heal. While the material acknowledgments do not matter to me, the overall acknowledgment does. I wake up every day carrying grief, knowing that I cannot vocalize to my colleagues why I did not respond promptly to their requests not to go to campus or a national conference. They ask me where I have been for the past two years. My response was, "Taking care of me". Many might even be perplexed or frown upon the fact that we are writing this piece together, but I am of the nature that if we can love and do good with grief, why not share our life-affirming strategies?

You see, many amazing, famous women of color scholars have died prematurely as well. I do not want to be one of them. I will not let this job kill me because my life matters more than any amount of productivity. I have more to live for. I plan to live as long as possible so that your brother and you can ask me as many questions as needed as you go through the stages of life when your father is not physically present. I made a promise to your father a long time ago that if he died, I would watch over you two. I am a woman of her word and fully intend to keep that promise. I hope this gives you some comfort. What is breathing life back into you these days, kiddo?

Love,
Tía

Dear Tía,                                                                                                        17 October 2023

I just got done reading your article. As I read, tears rolled down my cheeks. My father's bipolar disorder is something that we all have wanted to forget about. I know that I do not have a lot of pleasant memories of Dad during one of his episodes. We do not talk about it because it breaks all our hearts. Dad's death was not as much of a surprise as it was an expectation. Through many studies, we have seen that many people with severe mental illness do not prosper through life. It is the saddest thing to say that Dad was another statistic.

In the article, you talk about how, by having a simple conversation with the man who yelled outside your window, you realize that mentally ill people are not crazy, just different. The hard truth is that those differences are not accepted by society. When the world presents mental illness with such negativity, it destroys a person's mental health. I know it got to my dad.

The most brutal truth for me to face is the one thing that worries me is that I will end up as my father did. I know I do not have the same struggles as he did, but I still am a lot like him and have good and bad ways. When people see me, they see my father and look at me in shock. My father had all the potential to do great things, but he did not believe in himself. He was an intelligent person who was told he had issues that destroyed him. Mental illness always comes with a negative connotation. Sadly, I believe that most people who are mentally struggling feed into the negativity. Society sees people's struggles as trouble. As humans, we tend to fear the things we do not understand, which takes a toll on the people suffering. My father did struggle, but he was not a lost cause. Everybody saw his potential, and I know they see the same potential in me.

I want to do things with my life. I do not want to struggle like my father did all those years. I saw how much he would work and how much he enjoyed seeing my brother and me. I also saw the built-up aggression he had from not being content. Dad was not content with his life; he was angry and resentful because of all his regret. I do not want to live with all that pain and sorrow. Mental health care is the only way to persevere through the pain, and that is what I hope to do.

Love,
Solano

## 6. An Offering on Death, Loss, and Well-Being to Academia

I found myself distraught as the tears dripped down my face, and I held my cell phone, looking for the last reminisce of my brother and me—our text messages. My partner is trying to soothe my anxiety and pain. The Verizon employee making every attempt to recover the messages through iCloud. I think to myself, why did I upgrade? I did not imagine that the impact of not having our last conversations would cut so deep. I yearn for Mother Earth to bring these exchanges back to me. I realize they are not meant for my eyes only. They are meant to be shared—an offering. (Nichole, September 2023)

Through our epistolary exchanges, we speak of two overarching themes: intergenerational collective healing and affirming death and grief as a life-making practice. In this section, we further develop these themes as an analysis and an offering to BIPOC communities and academia for those who have or are enduring death, grief, and yearn for a space to center well-being in.

### 6.1. Intergenerational Collective Healing

BIPOC grief is not viewed as a priority to collectively engage or heal from. As BIPOC communities, we have a right to name our grief as it can be used as a tool to subvert racism and trauma. Grief is not a linear process; it is an ongoing ebb and flow and requires a mending of Western mind/body fragmentation. We extend this conversation to address the paucity of BIPOC-centered death, grief, and well-being in academia. We place our bodymindspirits within this existence of grief and our participation in the anti-racist struggle towards intergenerational collective healing [6]. Lara [6] poignantly asks: "How do we sustain the courage to bridge the sacred and the profane in our work, the spiritual and political in our lives, and our mind with our body?" (p. 436). Our response to this question, through our epistolary exchanges, is that we use words as tools to transform our pain and work towards intergenerational collective healing.

Due to Richard's premature death and confrontation with mental health, my nephew and I recognize the emotional splintering of our bodymindspirits as documented in our epistolary exchanges. Solano states:

I feel more comfortable when I am sick because my appearance finally reflects how I feel. Now I have a reason to stay in bed, now I have a reason to ignore the world, and now I can give myself time to heal. Grief is being sick on the inside; if you don't give yourself enough time to rest, you will stay sick. I am tired of feeling sick, tía. I am tired of struggling to be happy in a world that has taken so much from me.

Solano equates grief with sickness; while he is sick, he is given the space and time to heal. While the ideal space to heal might not be found in sickness in Western medicine or psychotherapeutic practice, for us, sickness is a recognition that our bodymindspirits require feeling our emotions. Anzaldúa [40] reminds us that:

you've learned that delving more fully into your pain, anger, despair, depression will move you through them to the other side, where you can use their energy to heal. Depression is useful—it signals that you need to make changes in your life, it challenges your tendency to withdraw, it reminds you to take action. To reclaim body consciousness tienes que moverte—go for walks, salir a conocer mundo, engage with the world (p. 553).

For us, sickness acts as a mechanism to reconnect and resist the Western mind/body split towards healing our fragmented selves towards wholeness and balance [6]. In response to Solano, Nichole wrote:

Grief is an embodied experience with its own process of moving through the body on its own terms. Your perception of grief as a sickness is very insightful. As a society, we are not taught how to grieve publicly, especially for people your age . . . I feel grief in my body every day. I have gained weight since he passed because I am experiencing depression and anxiety. My therapist says I am high functioning, but in all reality, I have to function to keep my job. You see, my job only rewards those who are productive and put on a happy face while you are slowly—to use your words—"sick" inside . . . I will not let this job kill me because my life matters more than any amount of productivity.

Nichole documents how grief has separated emotion from embodiment as if the body is only the senses rather than how we process experience. We consider grief as a rupture in our everyday lived experiences that we must continually process as ongoing rather than overcoming. Anzaldúa [40] argues that healing is an organic process that acts as a form of ". . . survival, claiming space, [and] decolonizing . . . [Healing] is a coming together, [a] restoration from pain, and trauma to a place of spiritual wholeness" (p. 545). Our intergenerational healing, in this context, births what Anzaldúa calls conocimiento (a consciousness), which is a "holistic epistemology that incorporates self-reflection, imagination, intuition, sensory experiences, outward-directed action, and social-justice concerns" [41] (p. 10). Conocimento is accessed through "events, emotions, memories, dreams, and other elements of personal experience" [41] (p. 10). Taken together, our conocimiento, through the suturing of our bodymindspirits, has allowed us to expose, confront, and heal from traumatic experiences caused by intersecting forms of death, grief, and oppression in academia. Our intergenerational collective healing has become our bridge to discuss how institutions (e.g., health care, education, mental health) were not designed for or with BIPOC communities in mind.

### 6.2. Affirming Death and Grief as a Life-Making Practice

We do not wish to sweep our grief under the rug to be ignored or for others to feel too uncomfortable to sit with and affirm it. After all, the one thing all of us will experience in life is death and grief. We finished this article on what would be my brother's 40th birthday. Our gift to him is the refusal to let death destroy our memories of him and the memories we continue to make with him in our dreams. Death and grief changed our previous versions of ourselves. However, we have learned from our epistolary exchanges as active grievers of color that to write about our pain in relationship to death and grief

is a practice of non-attachment meant to be witnessed. To be attentive to our grief in the moment allows us to honor its presence in our lives.

Badruddoja [7] theorizes life-affirming epistemologies of grief as a process that embodies multiple modes of knowledge and knowing to subvert discourses of BIPOC communities to ask, "What manages to live (survive and thrive) in the ruinous and damaged landscape of BIPOC trauma?" (p. 620). Identifying slow violence and racial harm among BIPOC communities, which is accentuated by grief and death, is only a partial purview. There are also modes of joy, love, strength, and resilience that offer healing for well-being within BIPOC communities. To answer this question, in using a critical race feminista epistolary methodology, intimate exchanges were documented not only as trauma but as a space to affirm our experiences with death and grief. We began our epistolary exchanges with a prologue introducing two critical moments that active grievers of color will go through in remembering and honoring their loved ones. These moments often go unrecognized and happen in the shadows of the aftermath of their loved one's death. First, they (i.e., family or kin) write and bear witness to the obituary of their loved one. No one prepares you to write this piece of your familial archive, but it acts as a written memory captured in a specific time and place to examine the sense of relationship you had with your loved one, their presence, and what they gifted the world. Our prologue started with honoring who Richard was in this physical world by sharing his obituary, which I (Nichole) wrote. Second, they find unexpected familial archives while going through their loved one's possessions after they have passed on. As time progressed after Richard's death, we had to go through his possessions; while painful, it was also a time that goes undefined. It was an intimate moment for us to grieve by crying, laughing, and asking, "Now, what was he thinking?" It is a critical time because it is the purview of your loved ones' most intimate memories, possessions, and ways of being before their death. It is their last breath of memory.

We conclude that the more we understand how grievers of color can influence the development of life-affirming strategies, the better we can prepare academic institutions to create the optimal grief environment that focuses on well-being within this space. Institutions of higher education can best support grievers of color by implementing transparent and equitable bereavement policies. Bereavement policies in academic institutions tend to be at an unpaid level or days, which is problematic for BIPOC communities as they often do not come or benefit from generational wealth. Grievers of color in academic institutions often have to choose to sustain their only source of income at the cost of their emotional, mental and physical well-being. More often than not, bereavement policies only apply to immediate family members, which can be difficult for grievers of color who might have had other family members or kin as their family source. Colleagues, staff, and students who engage with grievers of color might consider grief among BIPOC communities not as a problem to be "tended to" or "fixed" but rather as something that can be acknowledged with care. Replacing statements such as "How are you today?" with "How can I best support you today?" or "I am so sorry for your loss" with "Sending you light and positive energy during these difficult times" demonstrates an ethic of care. Finally, grievers of color should consider that while not everyone will come to understand their grief, they are allowed to create boundaries and redirect conversations that do not serve a purpose in their healing processes. Most importantly, grievers of color working in academic institutions should have the ability to say "no" without feeling guilty to preserve their emotional, mental, or physical well-being. Overall, the undertaking of life-affirming strategies for grievers of color in academic institutions has the potential to counter systematic racial violence as it offers a humanizing alternative to harm caused by race, racism, and white supremacy.

**Author Contributions:** Conceptualization, N.M.G. and S.F.G.; methodology, N.M.G.; formal analysis, N.M.G.; writing—original draft preparation, N.M.G. and S.F.G.; writing—review and editing, N.M.G. All authors have read and agreed to the published version of the manuscript.

**Funding:** This research received no external funding.

**Institutional Review Board Statement:** Not applicable, ethical review and approval were waived for this study due to the main source of data being letter exchanges which were secondary data sources this presents no more than minimal risk of harm to subjects and involves no procedures for which written consent is normally required outside the research context.

**Informed Consent Statement:** Not applicable, informed consent was waived due to the nature of the study.

**Data Availability Statement:** No new data were created or analyzed in this study. Data sharing is not applicable to this article.

**Conflicts of Interest:** The authors declare no conflict of interest.

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
