# Peer review of "Grief (Work) Is Heart (Work): A Critical Race Feminista Epistolary Exchange as an Offering on Death, Grief, and Well-Being to Academia"

_education, doi:10.3390/educsci14010058_

Round 1
Reviewer 1 Report
Comments and Suggestions for Authors
This is not only very important research, that not only points out the relevance of recognising individual and collective grief in the context of structural uniqualities, but also presents an impressive methotological approach that shows the interrelations of personal grief, social discourse and politics (here concerning CRT scholarship in the US). Many thanks to the authors!
Author Response
Reviewer 1: Thanks for your thoughtful words and the promise this manuscript has for the field.
Reviewer 2 Report
Comments and Suggestions for Authors
I was truly my honor to read this beautifully written and personal manuscript. As a Black women academic grappling with my own grieving process right now, your work help put language around many of the things I have been feeling on my own journey. I look forward to seeing this work published and being able to reference it in the future. Toward better understanding your narratives, I offer the following commentary:
· It would be helpful to add the dates to your letters, to help the reader imagine the fullness of the exchange.
· I found myself asking, how were these messages delivered (handwritten, emails?)
· What did your tía/ sobrino relationship look like before this grieving experience and how did this shared grief change/ affect this relationship?
· In you discussion you discuss your feelings of rage… However, this was not as apparently to me as a reader. It would be helpful if you briefly paraphrase or quote some areas that you were describing rage in your epistle(s); to better guide the reader in your though processes.
· In the final section called: “An Offering on Death, Loss, and Well-being to Academia”, are you able to offer some suggestions on how administrators and colleagues can becoming more affirming. Of course, this is different for everyone, but can you share a bit form your perspective about what could have been done to affirm you on your journey?
· At the end you go into a bit more detail about your brother/ father’s passing. It may be helpful, however, to know this upfront, before reading the epistles.
Author Response
Reviewer 2: Thanks for your thoughtful review. We appreciate your time.
- It would be helpful to add the dates to your letters, to help the reader imagine the fullness of the exchange.
Response: Corrected to reflect the dates associated with the epistolary exchanges, please refer to pages 14-24
- I found myself asking, how were these messages delivered (handwritten, emails?)
Response: Excellent question, they were email exchanges. We added this in the methodology section, please refer to pages 9-13
- What did your tía/ sobrino relationship look like before this grieving experience and how did this shared grief change/ affect this relationship?
Response: We added this in the methodology section, please refer to pages 9-13
- In you discussion you discuss your feelings of rage… However, this was not as apparently to me as a reader. It would be helpful if you briefly paraphrase or quote some areas that you were describing rage in your epistle(s); to better guide the reader in your though processes.
Response: We rewrote the theoretical framing of the piece to reflect the rewritten analysis. It was disconnected, but I was sure to provide quotes to guide the reader through the process. Please refer to pages 9-13 and 24-29
- In the final section called: “An Offering on Death, Loss, and Well-being to Academia”, are you able to offer some suggestions on how administrators and colleagues can becoming more affirming. Of course, this is different for everyone, but can you share a bit form your perspective about what could have been done to affirm you on your journey?
Response: We offered what we are referring to as life-affirming strategies for grievers of color at the end of this article. These emerged from our personal experiences. Please refer to pages 28-29.
- At the end you go into a bit more detail about your brother/ father’s passing. It may be helpful, however, to know this upfront, before reading the epistles.
Response: Agreed, We added more to the methodology and a prologue section, please refer to pages 9-14

Reviewer 3 Report
Comments and Suggestions for Authors
In this manuscript, the Authors address the particularities of loss, grieving, and healing, as experienced by BIPOC families, through a critical race feminista epistolary methodology. The Authors situate their process of grieving and healing from a deeply personal loss within the structural injustices People of Color continue to grapple with in the context of institutional racism across contexts, including education. In the Authors’ own words, “We engage a critical race feminista epistolary methodology through our intergenerational epistolary exchanges by documenting our knowledge, skills, and abilities pretraining to death, grief, and well-being in academia and beyond.”
First and foremost, I would like to thank the Authors for their powerful work. The richness with which they have illuminated their loss, grieving processes, and healing processes through epistolary method is moving, incisive, and brilliant. The Authors stated purpose of asking the Reader to bear witness to the Authors’ grief, and accompanying emotions, while provoking the Reader to confront their own losses and possible complicities in suppressing, rather than honoring, the grief of BIPOC communities was accomplished.
I offer the following suggestions for the Authors’ consideration to support clarifying and amplifying the argument.
Consider providing a bit more framing on the front end in terms of critical race feminista and chicana feminism to facilitate Readers’ meaning-making of the epistolary exchange. The methodology is well-described; however, I wanted some more theoretical framing and key themes on the front end to guide my engagement with the letters and anchor the discussion at the end. As written, the discussion of Rage as Constructive Healing and Affirming Grief felt a bit disconnected given the front matter and the richness of the epistolary exchange.
I also encourage the Authors to not undervalue their offering or knowledge/skills/abilities in the discussion at the end of the manuscript. I appreciate the connection to bell hooks and Solórzano & Yosso. Would it serve the Authors’ purpose for the manuscript to provide more of their own analysis/theorizing lifted out of the epistolary exchange around Affirming Racial Grief for Constructive Healing. I don’t think what you are doing should be labeled as microaffirmations. The knowledge/skills/abilities you have engaged are significant and can be impactful for BIPOC Readers who are struggling with processes of honoring grief (and all the emotions that come with it) and constructive healing to live forward. This may not be your purpose, but I think you also have Discussion points for Institutional change—Institutions need to give people, particularly BIPOC people, space, time, and resources for grieving through changes in policy, practice, and culture. Institutions cannot heal but I hear you calling on them to create viable avenues for BIPOC faculty (and their families) to heal as an on-going commitment because Racial Grieving is necessary for personal losses as well as collective losses from oppression.
There is minor copyediting needed in terms of spelling and punctuation.
Author Response
Reviewer 3: Thanks for your thoughtful review. We appreciate your time.
Consider providing a bit more framing on the front end in terms of critical race feminista and chicana feminism to facilitate Readers’ meaning-making of the epistolary exchange. The methodology is well-described; however, I wanted some more theoretical framing and key themes on the front end to guide my engagement with the letters and anchor the discussion at the end. As written, the discussion of Rage as Constructive Healing and Affirming Grief felt a bit disconnected given the front matter and the richness of the epistolary exchange.
Response: We appreciated this comment very much. We rewrote the theoretical framing of the piece to reflect the rewritten analysis. It was disconnected, but I was sure to provide quotes to guide the reader through the process. Please refer to pages 9-13 and 24-29
I also encourage the Authors to not undervalue their offering or knowledge/skills/abilities in the discussion at the end of the manuscript. I appreciate the connection to bell hooks and Solórzano & Yosso. Would it serve the Authors’ purpose for the manuscript to provide more of their own analysis/theorizing lifted out of the epistolary exchange around Affirming Racial Grief for Constructive Healing. I don’t think what you are doing should be labeled as microaffirmations. The knowledge/skills/abilities you have engaged are significant and can be impactful for BIPOC Readers who are struggling with processes of honoring grief (and all the emotions that come with it) and constructive healing to live forward.
Response: We want to thank you for this affirmation. We wrote this section completely, please refer to pages 24-29.
This may not be your purpose, but I think you also have Discussion points for Institutional change—Institutions need to give people, particularly BIPOC people, space, time, and resources for grieving through changes in policy, practice, and culture. Institutions cannot heal but I hear you calling on them to create viable avenues for BIPOC faculty (and their families) to heal as an on-going commitment because Racial Grieving is necessary for personal losses as well as collective losses from oppression.
Response: We offered what we are referring to as life-affirming strategies for grievers of color at the end of this article. These emerged from our personal experiences. Please refer to pages 28-29.
